# Predicting Anticancer Drug Resistance Mediated by Mutations

**DOI:** 10.3390/ph15020136

**Published:** 2022-01-24

**Authors:** Yu-Feng Lin, Jia-Jun Liu, Yu-Jen Chang, Chin-Sheng Yu, Wei Yi, Hsien-Yuan Lane, Chih-Hao Lu

**Affiliations:** 1Department of Medical Laboratory Science and Biotechnology, Asia University, Taichung 41354, Taiwan; yflin@asia.edu.tw (Y.-F.L.); yiwyiw0918@gmail.com (W.Y.); 2The Ph.D. Program of Biotechnology and Biomedical Industry, China Medical University, Taichung 40402, Taiwan; u107305606@cmu.edu.tw (J.-J.L.); u107311201@cmu.edu.tw (Y.-J.C.); 3Department of Information Engineering and Computer Science, Feng Chia University, Taichung 40201, Taiwan; yucs@fcu.edu.tw; 4Graduate Institute of Biomedical Sciences, China Medical University, Taichung 40402, Taiwan; hylane@gmail.com; 5Department of Psychiatry, China Medical University Hospital, Taichung 40402, Taiwan; 6Brain Disease Research Center, China Medical University Hospital, Taichung 40402, Taiwan; 7Department of Medical Laboratory Science and Biotechnology, China Medical University, Taichung 40402, Taiwan

**Keywords:** cancer drug, drug resistance, single amino acid variation, protein structure, machine learning, feature selection, personalized therapeutics

## Abstract

Cancer drug resistance presents a challenge for precision medicine. Drug-resistant mutations are always emerging. In this study, we explored the relationship between drug-resistant mutations and drug resistance from the perspective of protein structure. By combining data from previously identified drug-resistant mutations and information of protein structure and function, we used machine learning-based methods to build models to predict cancer drug resistance mutations. The performance of our combined model achieved an accuracy of 86%, a Matthews correlation coefficient score of 0.57, and an F1 score of 0.66. We have constructed a fast, reliable method that predicts and investigates cancer drug resistance in a protein structure. Nonetheless, more information is needed concerning drug resistance and, in particular, clarification is needed about the relationships between the drug and the drug resistance mutations in proteins. Highly accurate predictions regarding drug resistance mutations can be helpful for developing new strategies with personalized cancer treatments. Our novel concept, which combines protein structure information, has the potential to elucidate physiological mechanisms of cancer drug resistance.

## 1. Introduction

One of the greatest challenges of this century is precision medicine, highlighted by the search for personalized cancer medicine. The expectation is that by analyzing tumors at the molecular level, scientists can design treatment that specifically adapts to specific molecular subgroups of tumors and even individual patient characteristics, greatly improving the therapeutic outcomes. In recent decades, targeted cancer therapy has been associated with significant improvements in survival rates, so has become one of the standard strategies for cancer treatment [1]. However, a problem with targeted drug therapy is the emergence of cancer drug resistance [2], as with chemotherapy [3,4]. Many studies have investigated the mechanisms of resistance to chemotherapy and their solutions [5,6,7,8,9,10,11,12,13,14]. The mechanisms of how cancer cells acquire targeted drug resistance are not clarified. Some patients develop resistance to targeted drug therapy post-treatment, possibly due to the occurrence of drug-resistant mutations [15]. For example, gefitinib and afatinib, epidermal growth factor receptor tyrosine kinase inhibitors (EGFR-TKIs) that have been developed as targeted therapies for use in patients with non-small cell lung cancer (NSCLC) who have EGFR-activating mutations, significantly improve survival rates in this patient population [16,17,18]. However, a recent case study has reported that a patient with NSCLC and the EGFR L858R mutation developed acquired resistance to gefitinib [19]. Gene sequencing of the blood revealed an EGFR kinase domain duplication (EGFR-KDD mutation) that has not previously been reported as an acquired mutation due to EGFR-TKI resistance in NSCLC [19]. The gradual development of acquired resistance within tumors is characterized by subpopulations of cells that may acquire or already possess the mutations that enable them to escape the effects of targeted therapies [20,21,22]. Adaptive responses of tumor cells to treatment promote a variety of mechanisms that lead to drug resistance, including changes in the tumor microenvironment, repair of DNA damage, changes in drug targets, alterations of signaling pathways, and changes in the pharmacology of cells [4,23]. Many experimental studies have revealed intrinsic or acquired resistance mechanisms that validate the use of strategic changes to treatment strategies, such as combination therapy, to overcome resistance to tumor treatment [24,25]. Notably, searching for successful treatment strategies requires multiple experimental conditions, cell lines and different time series modeling techniques, all of which are expensive and time-consuming to conduct with traditional hypothesis-driven experimental methods [26,27,28].

In recent years, the public release of data from large-scale drug screening programs [29,30,31,32] has helped to promote the development of precision oncology. These data have been applied by computational methods to identify putative drug response biomarkers [29,33] and have been used to develop predictive models to predict drug sensitivity [34]. The development of computational strategies for predicting drug responses is vital for guiding drug discovery and reducing the required amount of experimental work. Various machine learning models have been proposed that predict drug responses and enable the discovery of drug response biomarkers, such as the random forest algorithm [34,35,36,37], support vector machine (SVM) algorithms [38], neural networks [39] and Bayesian multitask multiple kernel learning [40,41]. Many methods have been developed to predict drug responses using analytical data based on signaling networks, cellular dynamics, and high-throughput data [42,43,44,45,46].

Evidence from the study of genomics has produced vital information. As proteins are responsible for coordinating biological processes of cells, understanding and analyzing the role of proteins in organisms is necessary for correct assessments of disease status [47]. Proteomes are much more complex than their corresponding genomes, as proteomes involve mechanisms such as alternative splicing and post-translational modifications. Correct biological functions of proteins depend upon them coiling and folding correctly into three-dimensional (3D) structures; gene mutations can lead to structural changes that influence protein functions [48,49]. In many types of gene mutations, most disease-related single amino acid variations (SAVs) occur in structurally or functionally important positions. SAV refers to an amino acid substitution caused by a genetic polymorphism. In extreme cases, nonsynonymous encoding variants alter protein sequences and thus change the entire protein structure or function. The unique physical and chemical properties of each kind of amino acid mean that the occurrence of mutations at different positions in the sequence affects protein conformation and function to varying degrees.

In this study, we explored the relationship between resistance mutations and cancer drug resistance from the perspective of protein structures and we built a reliable prediction model to distinguish which mutation(s) might be responsible for drug resistance. The data used to build prediction models were obtained from the Catalogue of Somatic Mutations in Cancer (COSMIC) database, the world’s largest expert-curated database of somatic mutations in human cancers [50]. These mutations are recorded in the literature as drug resistance mutations and include both intrinsic resistance (before treatment) and acquired resistance (after treatment). The prediction models were constructed by machine learning-based methods, including genetic algorithms and SVMs, to predict drug resistance mutations involving protein structures. The performance of our combined model demonstrated an accuracy of 86%, an MCC (Matthews correlation coefficient) of 0.57 and an F1 score of 0.66, indicating that mutations may lead to drug resistance by altering protein structures and their positions in 3D space.

## 2. Results

### 2.1. Performance Evaluation of the Training Set

This study employed the machine learning method to construct several different drug resistance SAV prediction (DRSP) models. Each model was subjected to the five-fold cross-validation technique to evaluate predictive performance. All prediction models were optimized using the MCC as the fitness function. Table 1 presents comparisons of predictive performances from the different prediction models. In our experiment, the prediction of the simulated complexes (${\mathrm{DRSP}}_{s}$) outperformed the prediction of the crystal complexes (${\mathrm{DRSP}}_{c}$). The training set from the simulated and crystal complexes was divided into subgroups based on the spatial distance between the SAVs and the drug. According to the study dataset, approximately 60% of drug-resistant SAVs were located within an 8 Å distance from the drug; this 8 Å space surrounding the drug was regarded as the drug-binding pocket. We therefore used a cut-off of 8 Å, then divided the model according to SAV location; within the 8 Å space surrounding the drug (${\mathrm{DRSP}}^{i}$, where “i” = interior), or beyond that space (${\mathrm{DRSP}}^{e}$, where “e” = exterior). The interior models (${\mathrm{DRSP}}_{c}^{i}$ and ${\mathrm{DRSP}}_{s}^{i}$) had higher sensitivity than the exterior models (${\mathrm{DRSP}}_{c}^{e}$ and ${\mathrm{DRSP}}_{s}^{e}$), although the exterior models exhibited higher specificity than the interior models. Thus, we speculated that combining the ${\mathrm{DRSP}}^{e}$ and ${\mathrm{DRSP}}^{i}$ models would yield superior performances for both prediction models (simulated and crystal complexes). The combined interior and exterior modeled complexes yielded an accuracy of 85%, an MCC of 0.56 and F1 score of 0.65. The performance was slightly better with the combined interior and exterior crystal complexes, with an accuracy of 86%, an MCC of 0.57 and F1 score of 0.66, which may be due to the SAV positions, which are capable of directly or indirectly influencing drug binding to the protein. The superior predictive performance based on the 8 Å distance suggests that spatial distance is a primary consideration for calculations into direct and indirect factors affecting drug resistance. Indeed, spatial distance appears to be the primary contributor in the impact of resistance SAVs.

### 2.2. Performance Evaluation of the Testing Set

The testing set contained three resistance SAVs located in three proteins: L505H in the BRAF protein kinase (a participant in MAP kinase/ERK signaling) [51]; V215E in the dual specificity mitogen-activated protein kinase 2 (MAP2K2, a downstream kinase from BRAF in the MAPK pathway) [34]; and G2032R in the c-ros oncogene 1 (ROS1, a receptor tyrosine kinase that acts as an oncogene driver of non-small cell lung cancer [NSCLC]) [52] (Table 2). The molecular docking method was used to simulate the BRAF-vemurafenib, MAP2K2-PD0325901, and ROS1-crizotinib structural complexes. Different prediction models were used to perform drug-resistant SAV predictions, according to the distance between the drug-resistant SAV and the drug in the simulated complexes. Table 2 presents distances between drug-resistant SAVs and drugs, the prediction models, and the predicted results. The ${\mathrm{DRSP}}_{c}^{i}$ model correctly predicted three drug-resistant SAV mutations; L505H in BRAF, V215E in MAP2K2 and G2032R in ROS1. However, L505H in BRAF and V215E in MAP2K2 were not correctly predicted by the ${\mathrm{DRSP}}_{s}^{e}$ model. Specificity values of the testing sets predicted by the combined models from the crystal and simulated complexes were 0.84 and 0.93, respectively. The predicted results from the testing sets indicate that the crystal complex performs better than the simulated complex prediction model in the identification of drug-resistant SAVs. In contrast, the simulated complex prediction model performs better in detecting non-drug-resistant SAVs.

### 2.3. Case Study: L505 in BRAF and V215E in MAP2K2

The *BRAF* gene is an oncogene that occurs in 8% of human cancers [53], while the *BRAF* mutation usually occurs in melanoma and colorectal cancer [54,55]. The substitution of the valine residue by glutamic acid at amino acid position 600 of the BRAF protein is the most frequently observed *BRAF* mutation (occurring in approximately 90% of patients) [53,56]. Previous studies have documented how the use of a MAP2K2 inhibitor for *BRAF*-mutated melanoma triggers the PI3K/AKT pathway and leads to drug treatment resistance, although acquired *M**AP**2K2* mutations have also been associated with the drug resistance [57,58]. BRAF inhibitor therapy can provide short-term curative effects, but it also induces resistance and often disease progression [59]. Several unique drug-resistant SAVs have been reported in the COSMIC database. Five L505 mutations in the BRAF protein have been linked to vemurafenib; one of these L505 mutations is mutated to histidine [60,61]. Moreover, a V215E mutation in MAP2K2 is reportedly associated with PD0325901 drug resistance [62].

Our prediction model analyzes the interaction characteristics of drug-protein complexes. However, not all drug-protein complexes have a crystal structure for analysis. For those complexes lacking a crystal structure, we would use the docking model to simulate the complex before analyzing whether the SAVs are drug-resistant or not. Our docking model has proven reliable for our follow-up analysis, because the positions of each drug in the simulated complexes overlap the drug positions in the crystal complexes (Figure 1a, Figure 2a and Figure 3a). We were also able to simulate each mutated amino acid so that we could investigate changes in the structures after substitution (Figure 1b, Figure 2b and Figure 3b). Simulated mutated amino acids were generated by PS^2^, a protein structure prediction server [63]. The BRAF leucine residue 505 is 5.41 Å away from the docking space for vemurafenib, while the MAP2K2 valine residue 215 is 4.27Å away from PD0325901. Both drug-resistant SAVs are positioned within the distance of 8 Å. Thus, the ${\mathrm{DRSP}}_{c}^{i}$ would be applied. This model incorporates several critical features (see Appendix A): the drug-resistant SAVs are found with low levels of five elements (W-H-neu, W-Z-low, W-Z-med, W-P-low, and W-P-neu) within the surrounding amino acid residues. This finding implies that fewer neutral amino acids are associated with drug-resistant SAVs. BRAF residue L505 is positioned very near the functional amino acid R506, which exhibits a distinct conformation for molecular binding [64]. We speculated that this unique pattern can be used to predict drug resistance and highlight drug-protein interactions.

### 2.4. Case Study: The ROS1-G2032R Mutation

The rearrangement of the *ROS1* gene has been reported in several different tumors, including NSCLC [68]. Almost all patients with NSCLC receiving first- or second-generation tyrosine kinase inhibitors develop treatment resistance. The secondary kinase-domain mutation ROS1-G2032R has been identified in patients refractory to treatment with crizotinib, an ALK/ROS1/MET inhibitor [69,70]. The G2032R mutation also significantly reduces the cellular potency of lorlatinib [71]. Alteration of the glycine residue 2032 to arginine in the ROS1 structure is responsible for blocking the drug binding [72,73]. In our predicted system, the ROS1 glycine 2032 is 3.3 Å away from the docking space for crizotinib (within the 8 Å distance surrounding crizotinib), so it would be appropriate to apply the ${\mathrm{DRSP}}_{c}^{i}$ model. The amino acid residues surrounding drug-resistant SAVs are found with neutral and low polarizability properties. Glycine is simply an amino acid with a neutral charge, while mutated arginine is a large-volume amino acid with an electrically charged side chain. These massive changes in the microenvironment represent an obstacle for drug binding. Our prediction system predicted correctly in this case.

## 3. Discussion

A critical problem that has emerged with cancer-targeted therapies is persistent drug resistance, which is linked to various factors including increased drug efflux, epigenetics, inhibition of apoptosis, and drug inactivation [74,75]. These factors lead to escape pathways and adaptive mechanisms that can even include inactivation of the drug. Another challenge in drug resistance is the heterogeneity of tumors [76]. Genetic heterogeneity influences drug responses and causes drug resistance [77,78]. Although several large-scale tumor genomic projects exist, such as the 1000 Genomes Project [79], COSMIC [50] and the TCGA database [80], no models have systematically investigated the relationship between drug resistance and genetic mutations. Clarifying this relationship would benefit the drug development process, by serving as a reference during drug design and modifications. Our DRSP model attempts to provide a comprehensive high-throughput system for the full appreciation of drug resistance. Our approach offers a spatial structure that incorporates microenvironmental properties and interprets how drugs interact with protein residues. This system is advantageous for not only determining which SAVs are associated with drug resistance, but it also assists with drug development.

When we divided our prediction system into two subgroups, based on the 8 Å distance between docked drug and protein, we found that combining the interior (within the 8 Å distance) and exterior (beyond the 8 Å distance) models resulted in superior performances for both prediction models (simulated and crystal complexes). We suggest that this phenomenon may be related to the location of the SAV, indicating that it is closely associated with the drug interaction. The binding pocket is a dynamic, active site that hosts drug-protein interactions. Within this space, not all of the mutations are capable of influencing drug resistance; each amino residue within the 8 Å distance has a distinct function, such as arginine and histidine, which are more likely to be located in the ligand binding site [81,82]. More research is needed to explore these discrepant functions.

Our drug prediction systems analyzed selected features from the energy arising from interactions between drug molecules and SAVs, microenvironmental properties surrounding SAVs, protein structural characteristics and sequence conservation profiles of the SAVs (Figure 4). The four models share three features: the weighted contact number of aromatic amino acid, W-E-aro (FWY); the weighted contact number of sulfur amino acid, W-E-sul (CM), and an average entropy value containing 11 residues, ETP-avg11. Phenylalanine, tyrosine and tryptophan play a key role in the stability of the folding structure, while cysteine and methionine are important for structural maintenance; both molecules are hydrophobic. ETP-avg11 values may assist with evolutionary conservation.

Other features were selected for distances within 8 Å in the simulated and crystal complexes: the weighted contact number of the medium-volume amino acid, W-V-med (NVEQIL); the weighted contact number of the amino acid with neutral polarity, W-P-neu (PATGS); a BLOSUM62 substitution matrix, SSI-b62; and an average value over 7 residues, ETP-avg7. Around 40% of drug-resistant SAVs are located beyond the drug-binding pocket, but also lead to drug resistance. We also identified several features that are located beyond the 8 Å distance from the ligand: van der Waals forces, DKE-vdw; the weighted contact number of oxygen atoms, WCN-o; the weighted contact number of the amino acid with neutral polarity, W-H-neu (GASCTPHY); the weighted contact number of the small-volume amino acid, W-V-sma (GASCTPD); the weighted contact number of the amino acid with low polarizability, W-Z-low (GASDT); the weighted contact number of the acidic amino acid, W-E-aci (DE); a secondary structure, SSE; an H-bond acceptor of the backbone H-bond energy, EHB-acc; and an average value over 5 residues, ETP-avg5. Although further study is needed for most of these selected features to determine the mechanisms of drug resistance, our results indicate that it is possible to reliably predict drug-resistant mutations.

The greatest obstacle facing any attempt to reliably predict drug-resistant mutations is the identification of those SAVs capable of influencing drug resistance. The scant data in the literature makes it challenging to link with certainty any SAVs with drug resistance. Similarly, although several reliable prediction tools such as SIFT and PolyPhen can determine the pathogenic degree of variants, their rarity makes them difficult to define and their potential for drug resistance is uncertain. As illustrated by the TCGA records, only 132 BRAF missense variants (48 of which are predicted to be deleterious, i.e., associated with tumorigenesis, but are not necessarily drug-resistant) have been identified in 731 cancer patients across 26 tumor types; 82.76% of those patients have mutations in valine residue 600. The frequencies of other mutations amount to less than 2%, and most mutations are only detected once. Interestingly, both our crystal and simulated models predicted three mutations (F516L, M517V, and G596S) as capable of causing drug resistance (Figure 5a). Although the evidence is insufficient at this time, further research is warranted with these variants. The TCGA records have revealed the MAP2K2 protein in association with 51 missense variants in 57 patients across 16 tumor types; all of these variants are rare, as each appears only twice at the most. Moreover, 24 of these missense variants are deleterious. Interestingly, both of our prediction systems identified one of the 24 missense variants—mutation of G83 to serine was predicted as a drug-resistant mutation (Figure 5b). G83 is in a glycine-rich conservative loop, a catalytic region for binding and positioning ATP [83]. Although this mutation is not directly involved in drug interactions, it might alter drug binding and subsequently lead to drug resistance. TCGA data show that the ROS1 protein is found in 475 missense variants (40 of which have been identified by SIFT and PolyPhen as deleterious) in 384 patients across 29 cancer types; the frequencies of these variants are below 1%. One of those 40 missense variants (S2088 mutated to phenylalanine) was predicted to be a drug-resistant mutation by both of our systems (Figure 5c). Although most cancer-related mutations are rare, differences between individual patterns are important for precise medicine for tailoring targeting therapy.

Although the crystal drug-protein complex might more realistically reflect the binding condition, the crystallization structure has some limitations. Not all of the protein has the drug-protein co-crystal complex. Thus, we built the crystal complex and simulated two prediction systems to confirm the influence of SAVs on drug resistance. Our modeled systems can also analyze whether a particular SAV influences drug resistance without the crystal complex. Before performing any prediction, we would use the docking method to create the protein-drug complex. Although the conformation of the protein structure might differ slightly due to binding with different compounds, the binding site would be the identical region. A comparison of our stimulated and crystal complexes reveals consistent drug binding sites. Our stimulated complex is therefore reliable.

The primary impediment facing our study research is the small pool of data available for analysis. Although many large-scale cancer genome projects exist, scant information is available regarding drug resistance. Another challenge is how to confirm which SAVs would likely impact drug resistance. Understanding which SAVs influence the effects of a drug is invaluable for research and drug development in the cancer field. Our system provides characteristics of SAVs that help to elucidate their functions. The features used by our prediction systems may serve as valuable factors for further study.

## 4. Materials and Methods

### 4.1. Dataset Preparation

All SAV-related data were obtained from The Cancer Genome Atlas (TCGA) [80] and COSMIC databases [50]. The TCGA project has collected and sequenced gene mutations across many different cancers, while COSMIC is the largest available global database that includes somatic mutations from human cancers. To date, 582 unique drug-resistance mutations located in 22 genes have been annotated in COSMIC and 2,531 missense mutations located in these genes have been recorded in the TCGA. The impacts of these mutations can be appropriately interpreted by methodologies such as those used in studies predicting changes in amino acids that affect protein function, as with the SIFT (Sorting Intolerant from Tolerant) program [84] and by methodology that predicts the impact of protein sequence variants, such as PolyPhen [85]. For our study, non-drug-resistant SAVs were collected which were identified as deleterious by SIFT and PolyPhen. All SAVs were mapped to the identified protein structure obtained from the Protein Data Bank (PDB) via the Universal Protein Resource (UniProt). SAVs that lacked structure, were duplicated, or were ambiguous were filtered out, leaving a total of 136 drug-resistant and 589 non-drug-resistant SAVs in our dataset, which were classified into training and testing sets. The training set contained 11 protein-drug crystal complexes and included 133 drug-resistant and 477 non-drug-resistant SAVs. PDB alphanumeric identifiers (IDs) and the numbers of drug-resistant and non-drug-resistant SAVs are listed for each training set protein-drug complex in Table 3.

The testing set was used for the performance evaluation, which contained three drug-resistant SAVs (BRAF, MAP2K2, and ROS1) located in three proteins (Table 4). The BRAF-vemurafenib, MAP2K2-PD0325901, and ROS1-crizotinib complexes were simulated by the molecular docking method.

### 4.2. Construction of Prediction Systems

The machine learning method was used to build two DRSP systems. The first system contained two prediction models, ${\mathrm{DRSP}}_{c}$ and ${\mathrm{DRSP}}_{s}$, constructed according to the features generated by the crystal and simulated protein-drug complexes of the training set, respectively. The simulated protein-drug complexes of the training set were obtained by removing non-protein molecules from the protein structures and docking the drug into the protein structure. The second prediction system divided the training set into two subgroups, according to whether the minimum distance between drug and protein was greater or less than 8 Å in the protein-drug crystal complexes. Figure 6 shows the numbers of drug-resistant and non-drug-resistant SAV distributions for each distance interval (2 Å). A total of 60.09% drug-resistant SAVs were within 8 Å, so were regarded as located within the drug-binding pocket of proteins. These SAVs are considered to be capable of directly affecting the binding of the drug. The remaining 39.91% of drug-resistant SAVs were beyond the drug-binding pocket of proteins and were regarded as indirectly affecting drug binding activity. The training set was split into two subgroups (e and i), and four prediction models were built: ${\mathrm{DRSP}}_{c}^{e}$ and ${\mathrm{DRSP}}_{c}^{i}$ for the crystal protein-drug complexes and ${\mathrm{DRSP}}_{s}^{e}$ and ${\mathrm{DRSP}}_{s}^{i}$ for the simulated protein-drug complexes.

### 4.3. Machine Learning Method

Each prediction model was an SVM classifier module. The SVM method is a supervised learning model that uses statistical risk minimization to estimate the hyperplane of a classification. This model is widely used for classifying protein structure or function in computational biology [86,87,88,89,90,91]. All SVM calculations were performed using LIBSVM (version 3.24) [92,93], incorporating the radial basis function (RBF) kernel. The RBF kernel is the most generalized form of kernelization and is widely used because of its similarity to the Gaussian distribution. Optimized classification was identified by using hyperparameter tuning techniques for the given datasets. The parameters (penalty and gamma values of the RBF kernel) were trained by exponentially increasing the grid search from 2^−15^ to 2^15^, incorporating best values of informative measures with a 5-fold cross-validation during model training.

### 4.4. Feature Selection

The genetic algorithm (GA) [94,95,96] was used to select critical features and optimize classification performance. The basic GA procedures were as follows: N solutions ($S_{i},\;i=1,\ldots ,N$) were randomly generated in the initial population, with each solution $S_{i}\;$represented as a set of m-dimensional feature vectors $(f_{j}^{i},\;j=1,\ldots ,m$) indicating the binary representations of m features. If $f_{j}^{i}=1$, the $j^{th}$ feature was retained; if $\;f_{j}^{i}=0$, the feature $j^{th}\;$ was eliminated for feeding into the SVM. For each generation of τ, the three basic mechanisms driving the evolutionary processes were performed, consisting of the selection, mutation and crossover processes. The selection operators were defined as $\alpha^{\tau }=\max\left\{S_{1}^{\tau },\ldots ,S_{N/2}^{\tau },\alpha^{\tau -1}\right\}$ and $\beta^{\tau }=\max\left\{S_{\frac{N}{2}+1}^{\tau },\ldots ,S_{N}^{\tau },\beta^{\tau -1}\right\}$. The solutions $\alpha^{\tau }$ and $\beta^{\tau }$ had the best fitness values in each half of the N solutions and $\alpha^{\tau -1}$ and $\beta^{\tau -1}$ in previous generations, respectively. For the special case of τ=0, the fitness values of $\alpha^{0}\;$and $\beta^{0}\;$were defined as 0. A new solution in next-generation τ+1, $S_{i}^{\tau +1}$, was considered equal to $\alpha^{\tau }\;$if i was an odd numerical value, while $S_{i}^{\tau +1}\;$was equal to $\beta^{\tau }$ if i was an even numerical value.

Four informative measures (Equations (1)–(4)) calculated from the 5-fold cross-validation were used as the fitness functions in the selection process. They consisted of accuracy (Acc), the MCC, and the F1 score (F1), as well as the summation of sensitivity and weighted specificity (Hybrid) values, and were calculated as follows:(1)$$Acc=\frac{TP+TN}{TP+TN+FP+FN}$$
(2)$$MCC=\frac{TP\times TN-FP\times FN}{\sqrt{\left(TP+FP\right)\left(TP+FN\right)\left(TN+FP\right)\left(TN+FN\right)}}$$
(3)$$F1=\frac{2\times Precision\times Sensitivity}{Precision+Sensitivity}$$
(4)Hybrid=Sensitivity+δ×Specificity
where $Precision=\frac{TP}{TP+FP}$, $Sensitivity=\frac{TP}{TP+FN}$, and $Specificity=\frac{TN}{TN+FP}$, *TP* represents true-positives, *TN* represents true-negatives, *FP* represents false-positives, *FN* represents false-negatives and δ is the ratio of the number of positives to negatives.

After inputting the selection operators, two types of mutations were applied to the N solution $S_{i}$s. In the case of i=1,…,N/2, every b bit of the vectors was subject to mutation: b=~b, if the mutation rate was less than 0.1. In the case of $i=\frac{N}{2}+1,\ldots ,N$, we randomly chose a bit from each vector and subjected the bits to mutation without any mutation thresholds. The one-point crossover operations were carried out between $S_{2p-1}$ and $S_{2p}$, where p=1,…,N/2 and proceeded as follows: the feature vectors from r to m of $S_{2p-1}$ and $S_{2p}$ were swapped if the crossover rate was less than 0.5, where r was randomly selected from 1 to m.

In this work, the processes of feature selections and model training were repeated for a total of 8 runs. For each run, the number of generations was 150 and there were 80 solutions$\;\left(N=80\right)$ for each generation. Finally, the best feature set and predictive performances were selected with their optimized MCC values. The prediction system is shown schematically in Figure 7.

### 4.5. Generation of Feature Sets

In this study, we used 45 features to describe SAV characteristics, which were classified into four categories: the interaction energy between the drug and the SAV; the microenvironmental properties surrounding the SAV; the protein structural characteristics of the SAV; and the sequence conservation of the SAV (Appendix A). Three types of the interaction energies between drugs and SAVs (electrostatic, hydrogen-bonding, and van der Waals forces) were calculated from the crystal or simulated protein-drug complexes using iGEMDOCK software [97], a widely used molecular docking program that provides accurate predictions of protein-molecule interactions. Using iGEMDOCK with an accurate set of parameters ensured that the drugs were docked into specific proteins and the simulated protein-drug complexes were also obtained.

The weighted contact number (WCN) model [98] was used to describe our 26 microenvironment-associated features of SAVs. The local packing density profile of this WCN model is highly correlated with the sequence conservation profile [99]. The WCN value of atom *i* was calculated by WCNi=∑j≠iN1rij2, where r*_ij_* was the distance between atom *i* and atom *j*, while *N* was the number of calculated atoms. In this work, atom *i* was defined as the Cα atom of SAV, and the different microenvironment properties were represented by calculated different atom types or the source of atom *j*. The atom type of *j* could be Cα atoms, nitrogen atoms or oxygen atoms of an amino acid, representing the residue-, nitrogen- and oxygen-packing densities of a SAV, respectively. The packing density of the SAVs were then divided into different classifications representing the microenvironmental properties where the SAV was located. According to the physicochemical properties of the residues, we used the following classification schemes [100] to refer to the different microenvironments surrounding the amino acid residues: H for polar (RKEDQN), neutral (GASTPHY), and hydrophobic (CVLIMFW); V for small (GASCTPD), medium (NVEQIL), and large (MHKFRYW); Z for low (GASDT), medium (CPNVEQIL), and high polarizability (KMHFRYW); P for low (LIFWCMVY), neutral (PATGS), and high polarity (HQRKNED); F for charged (DEHKR), polar (CGNQSTY), and nonpolar (AFILMPVW); and E for acidic (DE), basic (HKR), aromatic (FWY), amide (NQ), small hydroxyl (ST), sulfur-containing (CM), aliphatic 1 (AGP), and aliphatic 2 (ILV).

Five structure-associated features were derived from the PDB and DSSP databases [101,102]. Firstly, the B-factor value of the SAV Cα atom was identified; this value represents the diminished intensity of scattered X-rays after atoms are displaced from their mean positions in a crystal structure. This displacement may be the result of temperature-dependent atomic vibrations, or because of static disorder in a crystal lattice. Our model also used critical DSSP information regarding solvent accessibility and energy derived from the acceptor and donor backbone hydrogen bonds. The last structural descriptors represent the secondary structure elements of SAV defined by DSSP; 1 for the α-helix (H, G and I), -1 for the β-sheet (B and E) and 0 for loop (T, S, and others).

In the last category of characteristics, 11 sequence-associated features were used in this study. The three commonly used substitution indices were used; the BLOSUM62 [103,104], PAM250 [105], and the position-specific scoring matrix (PSSM), which was derived from PSI-BLAST [106]. The evolutional entropy values derived from PSI-BLAST were used to denote a sliding window containing several amino acids on either side of the SAV. This window of amino acids was used to calculate average entropy values. Window lengths of 1, 3, 5, 7, 9, 11, 13 and 15 are centered on the SAV, representing sequence conservations from near to far-ranging.

## 5. Conclusions

In this project, we developed a drug-resistant prediction system that provides SAV environmental properties comprising the interaction energy, structure characteristics, and microenvironmental components. Although drug resistance is a more complicated mechanism involving more than a single mutation, an in-depth evaluation of the SAVs can determine how a particular SAV is associated with drug resistance and its value for drug development. Moreover, although targeted therapy is the preferred strategy for cancer treatment, it needs to consider molecular heterogeneity for the cancer being treated and the particular mutation pattern. Our prediction system may also be applied to different mutation patterns. The information gleaned can be used to generate individually-specific treatment strategies for true precision medicine.

## Figures and Tables

**Figure 1 pharmaceuticals-15-00136-f001:**
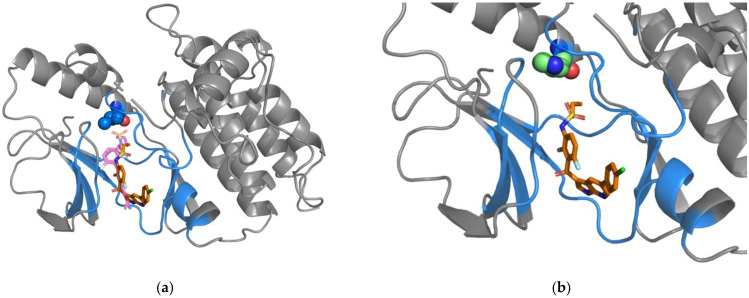
Simulated and crystal structures of the BRAF-drug complex. (**a**) The simulated structure of the BRAF-vemurafenib complex. Docked vemurafenib is indicated by the orange-colored stick. The magenta-colored stick represents the pyrazolopyridine inhibitor, which is located in the crystal structure of the BRAF-pyrazolopyridine inhibitor complex. The gray-colored cartoon structures of BRAF (PDBID: 3TV6 [65]) were drawn using PyMOL software. Residues found within 8 Å from vemurafenib are represented by blue coloring. The drug-resistant SAV (L505) is represented as spheres in blue. (**b**) Simulation of the amino acid mutated to histidine (H505) is shown in the color green.

**Figure 2 pharmaceuticals-15-00136-f002:**
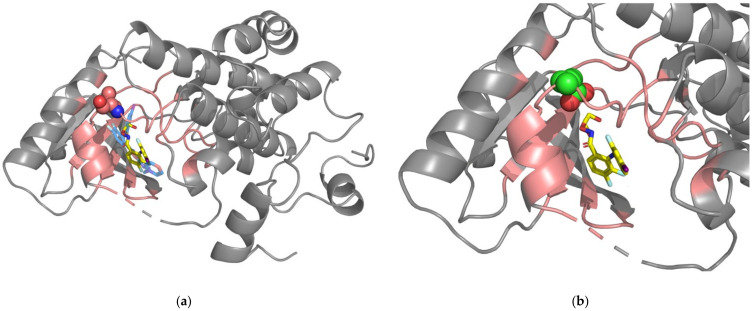
Simulated and crystal structures of the MAP2K2-drug complex. (**a**) The simulated structure of the MAP2K2-PD0325901 complex. Docked PD0325901 is indicated by the yellow-colored stick. The blue-colored stick indicates the PD184352-like inhibitor. The gray-colored cartoon structures of MAP2K2 (PDBID: 1S9I [66]) were drawn using PyMOL software. Residues found within 8 Å from PD0325901 are represented by pink coloring. The drug-resistant SAV (V215) is represented as spheres in pink. (**b**) Simulation of mutated glutamic acid (E215) is shown in the color green.

**Figure 3 pharmaceuticals-15-00136-f003:**
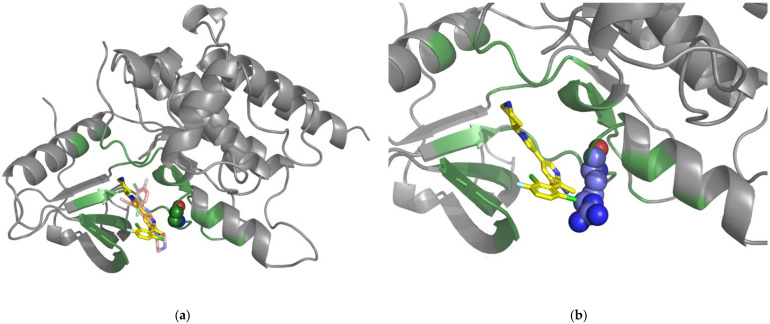
Simulated and crystal structures of the ROS1-drug complex. (**a**) The simulated structure of the ROS1-crizotinib structure complex. Docked crizotinib is indicated by the yellow-colored stick. The transparent orange-colored stick indicates crizotinib. The gray-colored cartoon structures of ROS1 (PDBID: 3ZBF [67]) were drawn using PyMOL software. Residues found within 8 Å from crizotinib are represented by green coloring. The drug-resistant SAV (G2032) is represented as spheres. (**b**) Simulation of mutated arginine (R2032) is shown in the color purple.

**Figure 4 pharmaceuticals-15-00136-f004:**
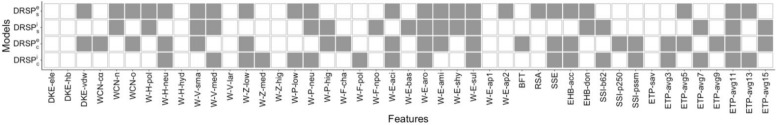
Selected features that were applied in the four drug prediction models.

**Figure 5 pharmaceuticals-15-00136-f005:**
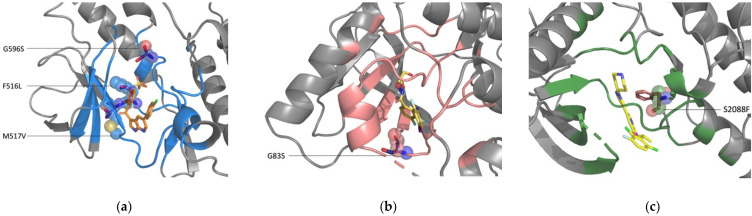
Simulated structures of the protein-drug complexes with mutation alterations. Wild-type amino acids are represented as spheres and mutated-type amino acids as sticks. (**a**) The BRAF-vemurafenib complex with the F516L, M517V, and G596S mutations. (**b**) The MAP2K2-PD0325901 structure complex with the G83S mutation. (**c**) The ROS1-crizotinib structure complex with the S2088F mutation.

**Figure 6 pharmaceuticals-15-00136-f006:**
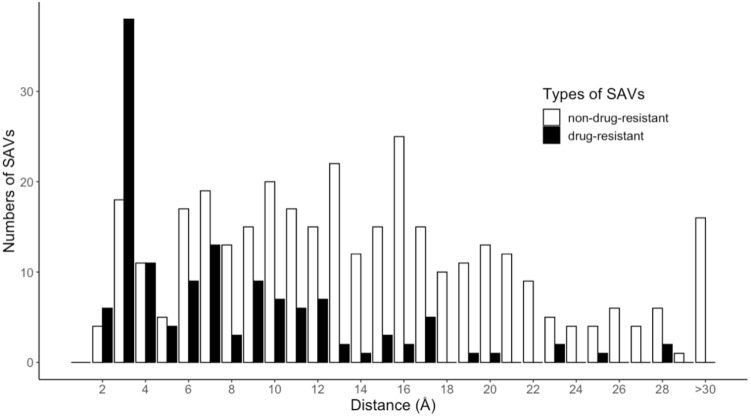
Distributions of drug-resistant and non-drug-resistant SAVs in the training set.

**Figure 7 pharmaceuticals-15-00136-f007:**
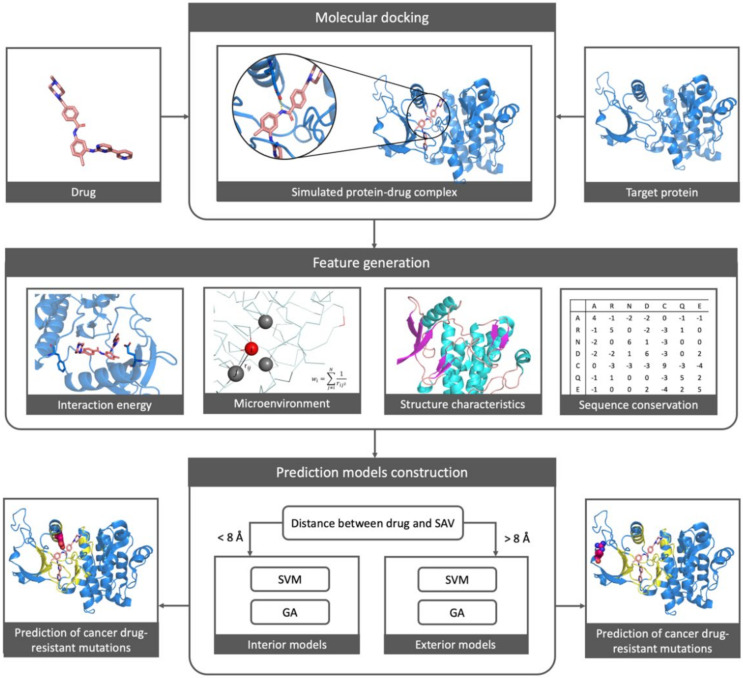
The workflow diagram represents the study’s prediction system for cancer drug resistance.

**Table 1 pharmaceuticals-15-00136-t001:** Comparison of predictive performances from different prediction models. All predictions were optimized using the Matthews correlation coefficient (MCC) as the fitness function.

Models	Accuracy	Sensitivity	Specificity	MCC	Precision	F1 Score
${\mathrm{DRSP}}_{c}$	0.8377	0.5338	0.9224	0.4936	0.6574	0.5892
${\mathrm{DRSP}}_{s}$	0.8508	0.5188	0.9434	0.5241	0.7188	0.6026
${\mathrm{DRSP}}_{c}^{e}$	0.8886	0.5385	0.9378	0.4803	0.5490	0.5437
${\mathrm{DRSP}}_{c}^{i}$	0.7819	0.7284	0.8224	0.5536	0.7564	0.7421
${\mathrm{DRSP}}_{s}^{e}$	0.8886	0.5577	0.9351	0.4888	0.5472	0.5524
${\mathrm{DRSP}}_{s}^{i}$	0.7660	0.6914	0.8224	0.5196	0.7467	0.7179
${\mathrm{DRSP}}_{c}^{e}+{\mathrm{DRSP}}_{c}^{i}$	0.8557	0.6541	0.9119	0.5724	0.6744	0.6641
${\mathrm{DRSP}}_{s}^{e}+{\mathrm{DRSP}}_{s}^{i}$	0.8508	0.6391	0.9099	0.5567	0.6641	0.6513

**Table 2 pharmaceuticals-15-00136-t002:** The predicted results from the testing sets.

Protein	Drug-Resistant SAV	Distance ^1^	Model	Predicted Result
BRAF	L505H	5.41	${\mathrm{DRSP}}_{c}^{i}$	TP
			${\mathrm{DRSP}}_{s}^{i}$	FN
MAP2K2	V215E	4.27	${\mathrm{DRSP}}_{c}^{i}$	TP
			${\mathrm{DRSP}}_{s}^{i}$	FN
ROS1	G2032R	3.30	${\mathrm{DRSP}}_{c}^{i}$	TP
			${\mathrm{DRSP}}_{s}^{i}$	TP

^1^ The distance between the drug-resistant SAV and the docked drug.

**Table 3 pharmaceuticals-15-00136-t003:** Numbers of drug-resistant and non-drug-resistant SAVs for each protein and drug in the training set.

Protein	Drug	PDB ID	Drug-Resistant ^1^	Non-Drug-Resistant ^2^
ABL1	Imatinib	1OPJ	31	36
ALK	Alectinib	3AOX	24	50
BTK	Ibrutinib	5P9I	4	36
EGFR	Osimertinib	4ZAU	15	54
ESR1	Raloxifene	1ERR	6	23
FLT3	Quizartinib	4RT7	5	48
KIT	Imatinib	1T46	21	51
MAP2K1	PD0325901	3VVH	2	31
PDGFRA	Sunitinib	6JOK	1	65
SMO	Vismodegib	5L7I	17	42
MET	Crizotinib	2WGJ	7	41
TOTAL			133	477

^1^ Numbers of drug-resistant SAVs; ^2^ Numbers of non-drug-resistant SAVs.

**Table 4 pharmaceuticals-15-00136-t004:** Numbers of drug-resistant and deleterious SAVs for each protein and drug in the testing set.

Protein	Drug	PDB ID	Drug-Resistant ^1^	Non-Drug-Resistant ^2^
BRAF	Vemurafenib	3TV6	1	48
MAP2K2	PD0325901	1S9I	1	24
ROS1	Crizotinib	3ZBF	1	40
TOTAL			3	112

^1^ Numbers of drug-resistant SAVs; ^2^ Numbers of non-drug-resistant SAVs.

## Data Availability

The authors confirm that the data supporting the findings of this study are available within the article and its Appendix A.

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
