# Peer review of "Predicting Anticancer Drug Resistance Mediated by Mutations"

_pharmaceuticals, 2022, doi:10.3390/ph15020136_

Round 1

Reviewer 1 Report

This is an interesting and novel manuscript evaluating role of mutations in chemoresistance development. A high quality schematic figure should be added to improve quality of article. There is no reference from 2020 and only two references from 2021. The first paragraph of introduction is about drug resistance; However, it does not provide full description of drug resistance. For instance, by understanding mutations, we can provide novel therapeutics for reversing chemoresistance. However, recent experiments have shown that nanoparticles can also prevent drug resistance. Please provide more statements about drug resistance. Suggested articles (Doi, 10.1016/j.carbpol.2021.118491; Doi, 10.1016/j.drudis.2021.09.020). The discussion section should be extended and elaborated. Add more descriptions to improve it.

Reviewer 2 Report

The authors presented an interesting study on the prediction of drug-resistane mutations. The study is well designed and conducted; however, in my opinion, it lacks tackling several aspects.

First of all, there are no references to similar studies of this type. In addition, the changes in ligand-interaction patterns upon mutation should be presented, visualized and compared. Moreover, the interpretation of the results should be more deeply discussed, as I understand, that actually the predictions are made on the basis of predictions (SIFT and Polyphen output)? 

Round 2

Reviewer 2 Report

The authors addressed all the comments raised in the review. The paper canbe published in the current form.